# Improving the Path to Obtain Spectroscopic Parameters for the PI3K—(Platinum Complex) System: Theoretical Evidences for Using [195]Pt NMR as a Probe

**Taináh M. R. Santos [1], Gustavo A. Andolpho [1], Camila A. Tavares [1], Mateus A. Gonçalves [1] and Teodorico C. Ramalho [1,2,*]**

[1] Laboratory of Molecular Modelling, Department of Chemistry, Federal University of Lavras, Lavras, MG 37200-000, Brazil

[2] Department of Chemistry, Faculty of Science, University of Hradec Králové, 500 03 Hradec Králové, Czech Republic

[*] Correspondence: teo@ufla.br

**Abstract:** The absence of adequate force field (FF) parameters to describe certain metallic complexes makes new and deeper analyses impossible. In this context, after a group of researchers developed and validated an AMBER FF for a platinum complex (PC) conjugated with AHBT, new possibilities emerged. Thus, in this work, we propose an improved path to obtain NMR spectroscopic parameters, starting from a specific FF for PC, allowing to obtain more reliable information and a longer simulation time. Initially, a docking study was carried out between a PC and PI3K enzyme, aiming to find the most favorable orientation and, from this pose, to carry out a simulation of classical molecular dynamics (MD) with an explicit solvent and simulation time of 50 ns. To explore a new PC environment, a second MD simulation was performed only between the complex and water molecules, under the same conditions as the first MD. After the results of the two MDs, we proposed strategies to select the best amino acid residues (first MD) and water molecules (second MD) through the analyses of hydrogen bonds and minimum distance distribution functions (MDDFs), respectively. In addition, we also selected the best frames from the two MDs through the OWSCA algorithm. From these resources, it was possible to reduce the amount and computational cost of subsequent quantum calculations. Thus, we performed NMR calculations in two chemical environments, enzymatic and aqueous, with theory level GIAO–PBEPBE/NMR-DKH. So, from a strategic path, we were able to obtain more reliable chemical shifts and, therefore, propose safer spectroscopic probes, showing a large difference between the values of chemical shifts in the enzymatic and aqueous environments.

**Keywords:** platinum complex; spectroscopic probes; PI3K; NMR; molecular dynamics

## 1. Introduction

Early diagnosis is the best chance for a positive outcome in the fight against cancer [1] and the search for precise spectroscopic probes that are very well-founded in their performance is increasing. Therefore, for an analysis involving a ligand that acts as a probe of a biological target, computational simulations are extremely welcome [2,3].

In this context, the PI3K enzyme is known for its association with several diseases, including breast cancer, making such an enzyme an important molecular target for the therapeutics of this disease. Thus, its inhibition becomes a promising strategy to combat this type of disease [4].

With that in mind, the search for molecules that act as spectroscopic probes for this enzyme becomes relevant. In that sense, the platinum complex (PC) shown in Figure 1 has the ability to inhibit the PI3K enzyme due to the presence of the benzothiazole-derived compound (AHBT) in its structural composition [5].

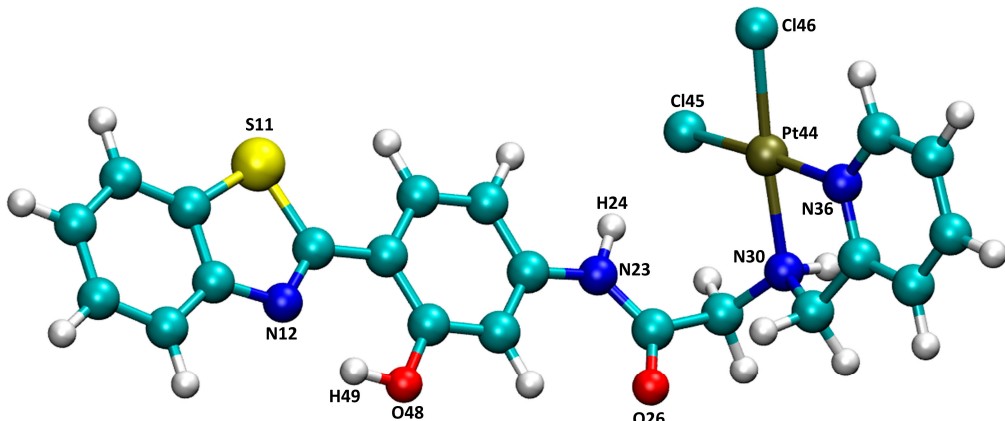

**Figure 1.** Three-dimensional structure of the *cis*-dichloro(2-aminomethylpyridine) platinum(II) bonded to 2-(4′-amino2′-hydroxyphenyl)benzothiazole (AHBT).

A platinum complex analogous to the one shown in Figure 1, studied in the work by Pereira et al. (2019) [3], was used as a spectroscopic probe for the PI3K enzyme. In [3], the benzothiazole derivative present in the platinum complex is the ABT compound, i.e., without the O48-H49 bond.

In the work mentioned above, a simulation of molecular dynamics was performed so that, subsequently, NMR calculations could be carried out. However, due to the lack of efficient force field parameters to describe the platinum complex, a quantum molecular dynamics simulation was conducted with a very short calculation time. This makes it impossible for a deeper analysis to take place.

A new possibility arose when, in 2021, another group of researchers developed a new and efficient AMBER force field specifically for PC (Figure 1) [6]. From there, we propose to use this force field (FF) to perform a simulation of classical molecular dynamics (MD) between the PC and PI3K at the nanosecond scale, allowing a deeper investigation to occur.

Thus, we initially carried out a docking study to obtain the best orientation and perform an MD simulation (50 ns) with an explicit solvent. Soon after, a second MD (50 ns) was performed only between the PC and the explicit water molecules. Finally, NMR calculations in an enzymatic environment and in an aqueous environment were conducted.

The two MD simulations allowed the selection of the best amino acid and water residues through the analysis of hydrogen bonds and Minimum Distance Distribution Functions (MDDF) [7], respectively. Furthermore, to select the most important frames of the MDs, the sophisticated mathematical resource, OWSCA [8], was used.

Based on this, more precise NMR calculations (i.e., with strategically determined frames and residues) allowed the computational cost of quantum calculations to be reduced and the obtaining of spectroscopic parameters to be improved. As a consequence, a more reliable study on the performance of the PC with PI3K is proposed, providing better and safer spectroscopic probes.

## 2. Methodology

### 2.1. Molecular Docking

The starting coordinates of PC for the docking simulation, besides the RESP atomic charges, were obtained from the work of Pereira et al. (2021) [6], where the B3LYP functional and the LANL2DZ ECP basis set were used for the Pt atom and def2-TZVPP for the other atoms.

The crystallographic structure of PI3K and the active ligand (*N*-(6-[2-(methylsulfanyl)pyrimidin-4-yl]-1,3-benzothiazol-2-yl) acetamide) docked in the active site of the PI3K are provided from the work of D'Angelo and collaborators (PDB code: 3QJZ) [9]. Thus, the position of the active ligand served as a guide to the PI3K active site region where PC was docked. The simulation was performed considering a constraint of 11 Å and flexible

residues at the same distance (totaling 155 flexible residues). In addition, the resolution of the grid used was 0.30 Å. A total of 100 poses were requested in the simulation using the Molegro Virtual Docker software [10].

The poses obtained from the docking study were analyzed so that only one could be selected. Among the various possible orientations of the PC in the PI3K active site, we selected the conformation that presented a structure orientation closer to the crystallographic structure of the active ligand of PI3K (*N*-(6-[2-(methylsulfanyl)pyrimidin-4-yl]-1,3-benzothiazol-2-yl) acetamide) (Figure 2). Furthermore, another relevant criterion was reproducing the interactions already reported in the literature with favorable interaction energy (H-Bonds between PC and residue Val882). In this sense, the most favorable orientation was prepared for the next step, the classical molecular dynamics simulation.

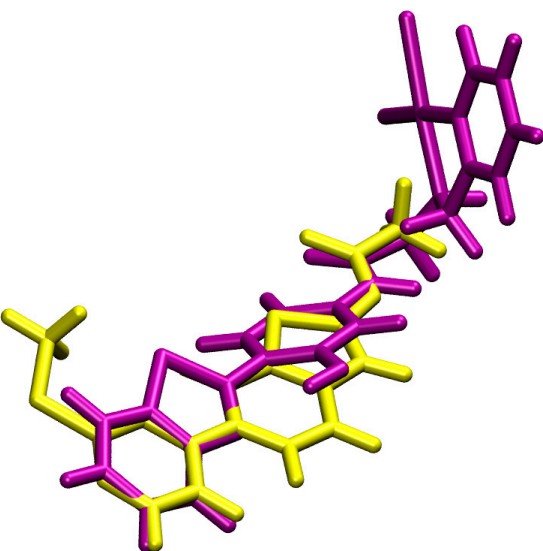

**Figure 2.** Overlapping of the selected pose (purple) and the PI3K active ligand (*N*-(6-[2-(methylsulfanyl)pyrimidin-4-yl]-1,3-benzothiazol-2-yl) acetamide) (yellow).

### *2.2. Classical Molecular Dynamics*

The new and efficient AMBER force field developed in [6], specifically for PC, was used to describe the complex. For the enzyme, the leaprc.ff14SB force field was used, in addition to the TIP3P model, to describe water molecules [11,12]. Eight $Na^+$ ions were added to neutralize the charge of the system.

Initially, a minimization of the system occurred to relax the system and remove the bad contacts between the atoms. Soon after, a heating ramp from 0 to 310 K was performed and, subsequently, the system was equilibrated at 310 K. The last step, production, was performed with NPT *ensemble*, in 50 ns generating 2000 frames.

As a way of comparison with the simulation of PC within the active site of PI3K, another simulation was performed, this time with the complex without the enzyme, only in an aqueous environment. This second MD followed the same procedure described in the previous lines. In both MD simulations, the AMBER 20 software [13] was used. From these two simulations, subsequent NMR calculations were analyzed considering the enzymatic environment and the aqueous environment.

### 2.2.1. Selection of the Best Snapshots and Residues

The best frames of the trajectories of the two MDs were selected with the assistance of an efficient mathematical tool, the OWSCA algorithm [8]. The selection of conformations by the OWSCA method was performed using the Wavepress program [8].

OWSCA is based on the wavelet analysis, which is a mathematical procedure that converts a sign in a different form. Thus, the discrete wavelet can be defined according to Equation (1),

$$d_{j,k} = \int_{-\infty}^{\infty} x(t)\psi_{j,\,k}\, k(t)dt \qquad (1)$$

where $d_{j,k}$ is the wavelet coefficient, $t$ is the variable time (normalized between 0 and 1), $j$ represents the scaling parameter (resolution), which determines the time and frequency resolutions of the scaled wavelet function $\psi$, and $k$ represents the shifting parameter, which translates the scaled wavelet along the time axis. In the present work, we use the Haar wavelet for the selection of structures. The Haar wavelet can be defined according to Equation (2).

$$\psi_{Haar(t)} = \begin{cases} -1 & 0 \leq t < 0.5 \\ 1 & 0.5 \leq t < 1 \\ 0 & Otherwise \end{cases} \qquad (2)$$

In addition, aiming to increasingly reduce the computational cost of NMR calculations, the best residues of amino acids (first MD) and water (second MD) were selected. The selection criteria followed, respectively, the investigation of hydrogen bonds and the Minimum Distance Distribution Function (MDDF) [7].

### 2.3. Nuclear Magnetic Resonance (NMR) Calculations

From the strategic selection of the best frames and residues of the two MDs described above, the NMR calculations were performed. The software and method used in the calculation were, respectively, Gaussian 09 [14] and the GIAO-DFT method [15], with the functional PBEPBE [16] and the NMR-DKH basis set, designed to obtain NMR parameters for platinum compounds [17].

Once the shielding tensors were derived from these calculations, to obtain the $^{195}$Pt chemical shifts, the cisplatin (experimental chemical shift $\delta(^{195}\text{Pt})$ = −2097 ppm) was used as reference [17], and the same theory level described above applied to obtain cisplatin shielding tensors. This procedure was successfully used by Pereira et al., 2019 [3].

## 3. Results and Discussion

### 3.1. Docking

After analyzing the 100 poses obtained from the docking calculation, the best orientation was selected (Figure 3). As can be seen, three hydrogen bonds (H-Bonds) were identified involving the platinum complex and the valine (Val882) and glutamic acid (Glu880) amino acids. In addition, the pose interaction energy was −158.7 kcal/mol, suggesting that the PC interacts favorably in the PI3K cavity.

From the observation of Figure 3, it is possible to notice that the PC acts both as an electron donor and acceptor. In the first interaction between the complex and valine (2.31 Å), the PC acts as an electron donor (PC:H24···O:Val882). In the second H-Bond (2.56 Å), the PC accepts electrons from valine (Val882:H···O48:PC). Finally, in the last interaction shown in Figure 3 (2.96 Å), the PC acts as an electron donor for glutamic acid (PC:H49···O:Glu880).

As previously mentioned, different from the platinum complex studied by Pereira et. al. (2019) [3], in the complex under investigation in this work, there is an extra O48-H49 bond. For this reason, it was not possible to compare whether interactions involving O48-H49 were also found in [3]. On the other hand, as pointed out by Pereira and co-workers [3], the interaction between PC and Val882 was also found (PC:N23-H24···O:Val882) in our work.

The docking study is a static investigation, i.e., a time scale is not considered. In addition, the effect of temperature and the effect of the solvent are not considered. Thus, an analysis over time becomes suggestive, considering both the thermal effect and the solvent effect [18].

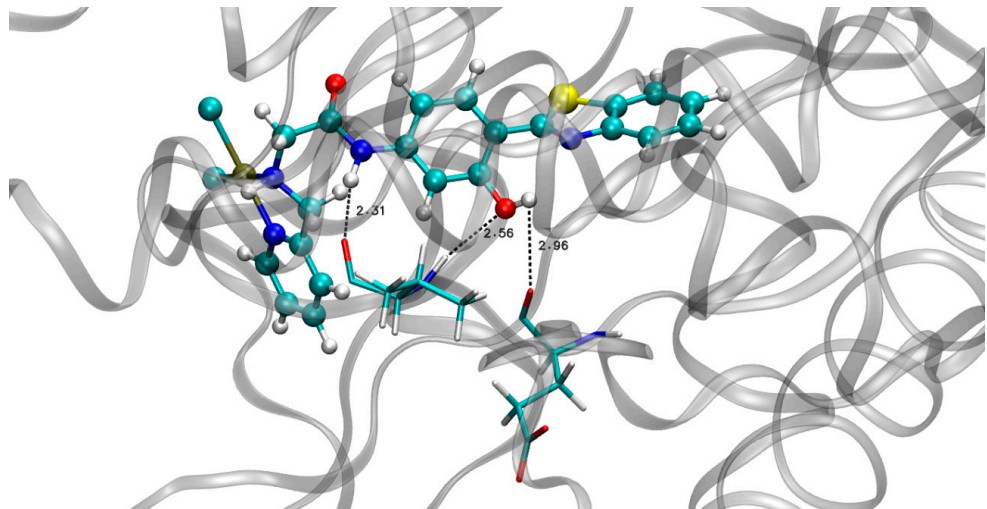

**Figure 3.** Best pose obtained by the docking study. At the bottom left, Val882 residue is forming two H-Bonds (2.31 Å and 2.56 Å) with PC. At the bottom right, an H-Bond (2.96 Å) can be seen between the Glu880 residue and PC.

### 3.2. MD in Enzymatic and Aqueous Environment

There are reports in the literature about the development of an FF for the cisplatin molecule. These studies began in 1994 through the work of Yao and collaborators [19], where they used statistical analysis to infer on bond stretching and bond angles parameters. This study was further refined later by the work of Scheeff and collaborators [20]. The cited works served as a starting point for the formulation of a new methodology for the parameterization of coordination compounds with platinum developed from quantum molecular dynamics calculations [21].

In this same perspective, it is possible to find, in the literature, works that present other parameterization strategies for OPLA-SS2 [22] and POSSIM [23] force fields [24]. However, in many cases, especially involving metal complexes, the developed parameters are not transferable among several complexes since each compound has its peculiarities. For these reasons, the absence of parameters to describe platinum complexes in MD simulations remains a major challenge. Thus, the development of a specific force field for the PC [6] paved the way for a reliable MD simulation, enabling the study of the PC with an important biological target, PI3K.

Based on the best pose stated in the docking simulation (Figure 3), an MD calculation (50 ns) was performed. Changes in the spatial coordinates of PC and PI3K along the entire trajectory are indicated using the RMSD graph (Figure 4). The x, y, and z coordinates of the first frame of the trajectory were used as a reference for the RMSD calculation.

From the analysis of Figure 4, it is possible to notice the excellent evolution of the PC over time, presenting an average of only 1.42 Å and standard deviation of 0.14 Å. The oscillation of the enzyme along the entire trajectory resulted in an average of 3.68 Å, a value considered low for a large system with numerous degrees of freedom. Furthermore, the standard deviation of its oscillation was only 0.65 Å, which shows that there were no large variations. However, to ensure that the system has in fact reached equilibrium, we considered the last 30 ns (20–50 ns) of the simulation for the analysis.

The analysis of the hydrogen bonds formed between the PC and PI3K was performed using the VMD (Visual Molecular Dynamics) software [25]. A cutoff radius of 3.5 Å and cutoff angle of 30° were considered. Two H-Bonds were found, both involving the amino acid valine (Val882), as shown in Figure 5.

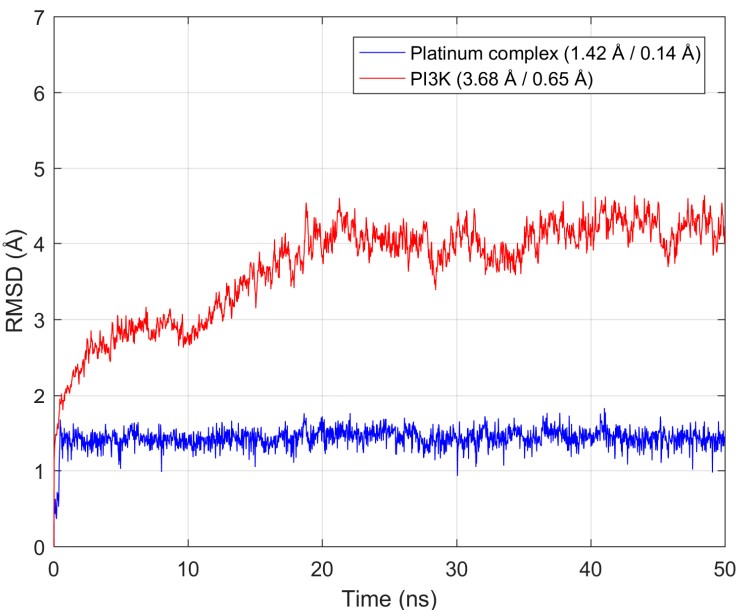

**Figure 4.** RMSD vs. Time. The behavior of PC is shown in blue and, in red, the evolution over time of the PI3K.

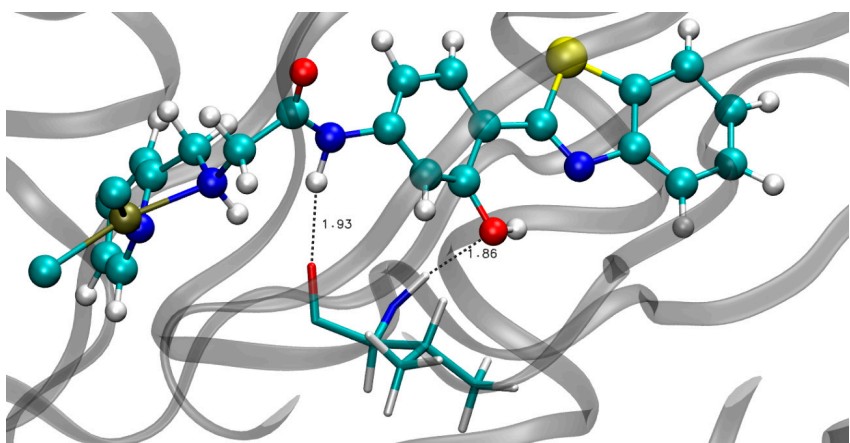

**Figure 5.** Two hydrogen bonds with higher frequency of occurrence between the PC and Val882 residue (1.93 Å and 1.86 Å) during the MD simulation.

In line with the work of Pereira et. al. [3] and with our docking study, the majority bond PC:H24⋯O:Val882 (1.93 Å) was found, where the PC acts as an electron donor. Furthermore, the PC received electrons from Val882 through the Val882:H⋯O48:PC bond (1.86 Å). These two H-Bonds had the highest frequencies of occurrence in the entire MD simulation with 87.25% and 69.50%, respectively.

In addition, we plotted the graph of the distances versus time of the two H-Bonds pointed out with the highest frequency of occurrence in the considered analysis time (20–50 ns). The first graph (Figure 6a) refers to the PC:H24⋯O:Val882 interaction, where the average bond length corresponds to 1.99 Å and the standard deviation was only ±0.28 Å. The second graph (Figure 6b) corresponds to the Val882:H⋯O48:PC hydrogen bond, with a mean distance of 2.24 Å and a standard deviation of ±0.22 Å.

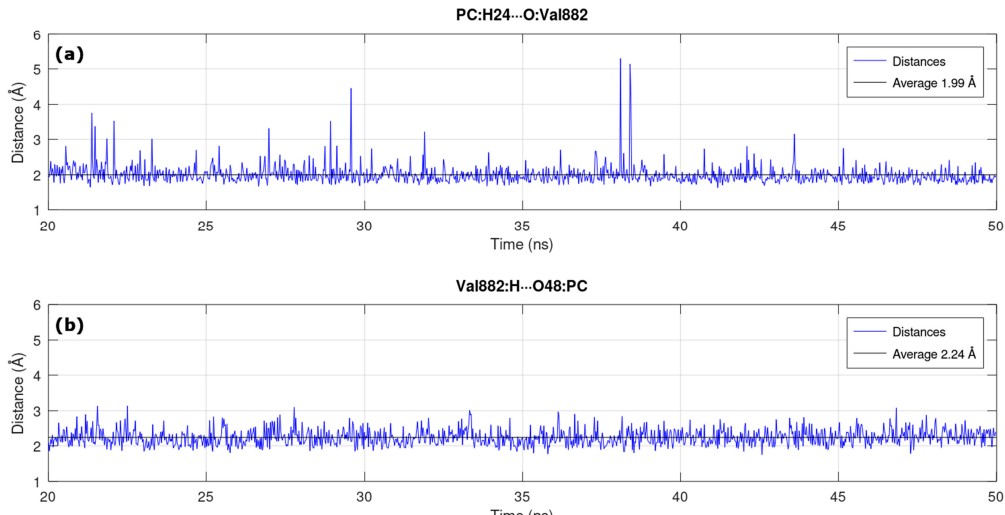

**Figure 6.** Bond lengths versus time of the two H-bonds with the highest frequency of occurrence in the MD simulation. (**a**) PC:H24···O:Val882; and (**b**) Val882:H···O48:PC.

On the other hand, the bond between the PC and glutamic acid (PC:H49···O:Glu880, 2.96 Å) shown in Figure 3 was not found in the MD simulation. This reflects the importance of complementing the docking simulation with the MD simulation, because interactions that are pointed out initially may not remain during a long simulation period. In this way, a deeper study is carried out with the combination of these two techniques.

In order to compare the results of the NMR calculations in an enzymatic environment and in an aqueous environment, a second MD simulation was carried out only between the PC and water molecules. This new calculation was performed following the same procedure presented in Section 2.2.

To select the best residues of water molecules, the Minimum Distance Distribution Functions (MDDF) [7] were used. The MDDFs are functions similar to the Radial Distribution Functions, however, the MDDFs consider the complete solute, not just the center of mass as the Radial Distribution Functions, which more efficiently characterizes how the solvent molecules are near the solute [7].

Thus, as can be seen in Figure 7, the total density of water molecules that can be found in a radius of 8 Å with respect to the PC is depicted in blue. Therefore, at short distances, there is a greater density, and as the distance increases, the function converges and stabilizes. The peaks presented refer to the solvation spheres.

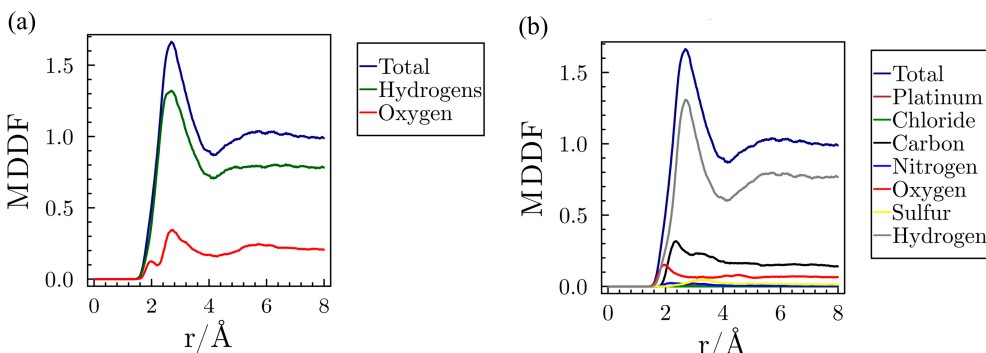

**Figure 7.** Minimum Distance Distribution Function of water with respect to PC. (**a**) Contribution of O (red) and H (green) atoms to the total function (blue). (**b**) Contribution of the PC atoms to the total function (blue).

In Figure 7a, the function in red represents the contribution of all the O atoms in the water molecules to the total function. The first peak characterizes the first sphere of solvation,

which is related to the specific interactions between the PC and the water molecules. The second peak represents the second solvation sphere and refers to non-specific interactions. The curve in green depicts the contribution of H atoms to the total function. The sum of the two curves (red and green) results in the function of total distribution.

Furthermore, from Figure 7b it can be seen that the hydrogen atoms contribute the most to the total function, whereas the Pt atom practically does not contribute to the total minimum distance distribution function (blue curve in Figure 7b).

Given the above, a radius of 3.2 Å was adopted to ensure that all interactions, specific and non-specific, were considered. In this way, all water molecules present at this distance would be included, along with PC, in the NMR calculation. Through this resource, it was possible to obtain a great decrease in the computational cost of quantum calculations (NMR).

### 3.3. Selection of the Best MD Structures—OWSCA

Aiming to reduce the amount of quantum calculations in this work, the best configurations of the last 30 ns of the two MDs were selected. For this, a sophisticated computational resource was used, the OWSCA algorithm. Indeed, this method is fully optimized and presents satisfactory results in the selection of structures [26].

As described in Section 2.2.1, we used the Haar wavelet, from the Daubechies family. The selection of the best wavelet is an important step since the wrong choice, i.e., selecting a wavelet from a family that does not properly select the frames from the MD simulation, can impact the statistical data derivate of the system and will affect all the next steps from the work, in our case, the quantum mechanics calculations [26].

It is worth mentioning that this methodology makes the selection of the frames by the compression of the total MD result, using the wavers coming from the system behavior. This is different to the traditional cluster selection, which groups the frames according to structural similarity. The advantage of this approach comes from the number of statistical values of the system that lead to the selected frames for the following calculations [26].

Keeping this in mind, Figure 8a,b shows the treatment of the signal obtained for MD simulations, the signal in blue represents the original configurations of the system, and the signal in red represents the configurations treated by the OWSCA methodology.

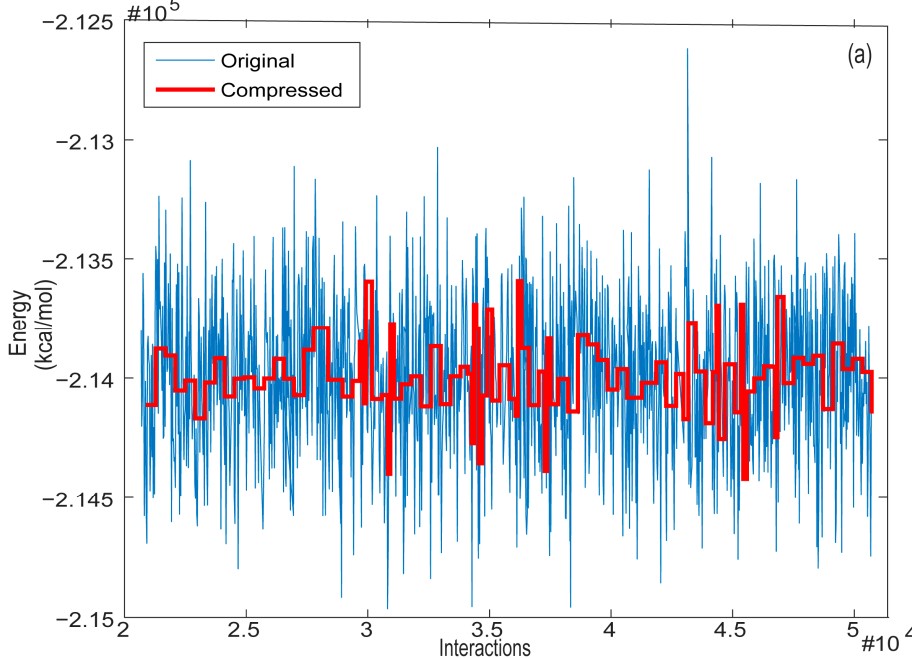

**Figure 8.** *Cont.*

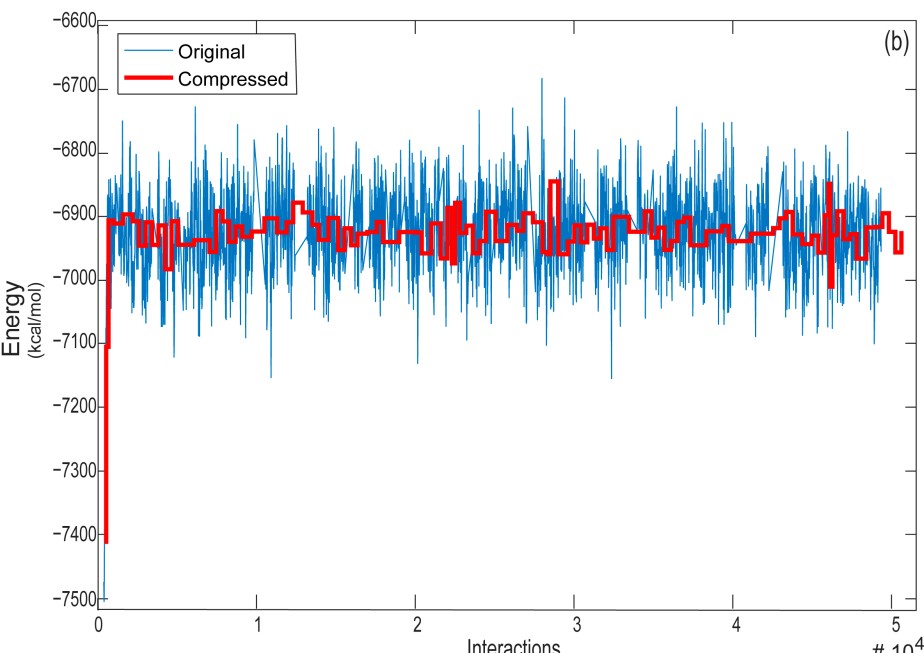

**Figure 8.** Energies of the systems studied (original and compressed) at each moment. (**a**) First MD with PC and PI3K; (**b**) Second MD with PC and water molecules.

The original signal obtained by MD simulations generated 2000 conformations. Thus, the treatment of the signal of the first system (PC plus PI3K, Figure 8a) resulted in the selection of 93 structures, whereas the treatment of the signal of the second system (PC plus water molecules, Figure 8b) resulted in the selection of 98 structures. It is important to mention that the treatment of both signals originated a small number of conformations, showing that the used method is efficient in reducing the number of structures obtained by MD simulations.

*3.4. NMR Spectroscopy*

The entire path taken so far was strategically designed to obtain better spectroscopic parameters. Starting from a specific parameterization for the platinum complex, we offer accurate analyses and more reliable results. In addition, a critical point of computer simulations is the computational cost.

In this way, we propose strategies to reduce the number of frames that would be submitted to NMR calculations, in addition to an appropriate selection of the main amino acid residues and water molecules.

In line with that, we proposed to reduce the PC:PI3K system to be able to perform GIAO-DFT calculations without a very high computational cost. For this reduction, other studies indicate that only the residues of the PI3K active site in the range for performance H-Bonds can be considered for studies of the complex–enzyme relationship [2,3,27]. In the PC:PI3K system, the reduced system is composed by the platinum complex and the residue Val882. The representation of the reduced system can be seen in Figure 9.

With this in mind, chemical shifts calculations were performed at a theory level PBEPBE/NMR-DKH, using the 93 and 98 frames selected with amino acid residues and water molecules, respectively. The average chemical shifts values for both environments were presented in Table 1. These calculations point out that, as expected, there is a difference when the chemical environment was changed from water to enzymatic of around 1600 ppm.

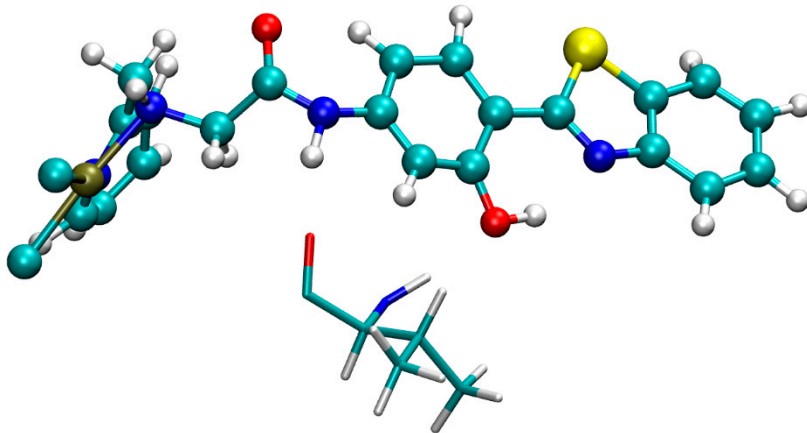

**Figure 9.** Reduced system of PC:PI3K used for NMR calculations, considering the platinum complex and the Val882 residue.

**Table 1.** Theoretical $^{195}$Pt chemical shifts in different chemical environments performed at theory level PBEPBE/NMR-DKH.

| Chemical Environment | $\delta(^{195}\text{Pt})$ (ppm) |
|:---:|:---:|
| Enzymatic | −1306.58 ± 25.33 |
| Aqueous | −2911.25 ± 9.66 |

This large difference between the chemical shift in enzymatic (−1306.58 ppm) and aqueous (−2911.25 ppm) environments is important to understanding the behavior of PC in these two environments.

Since our goal is to verify the difference in the chemical shift value in the different environments, it is important to verify if the mean results of the chemical shifts are statistically different from each other. For this we performed a Student's *t*-test, considering a confidence level of 95%, taking into account that the null hypothesis is that the two means are equal. For our data, the *p*-value was equal to 0.029, which leads us to rule out the null hypothesis, that is, the mean chemical shift values for the platinum complex in an aqueous and enzymatic environment are statistically different.

The higher chemical shift in the enzyme active site is an expected result, since the active sites have a hydrophobic characteristic, given that H-Bonds are formed between the PC and the PI3K enzyme, highlighting the importance of these interactions for changes in the chemical shift. The same behavior was observed for similar compounds, with Pt and Re, in previous studies [2,3].

These results indicate that the PC shows a different behavior in both environments, showing the sensitivity of the Pt chemical shift to these media, which is an interesting result for the application of this complex as a spectroscopic probe.

## 4. Conclusions

From the docking and MD (50 ns) studies between the PC and PI3K, we point out the Val882 residue as the most frequently occurring amino acid in the entire MD simulation. Thus, 93 frames were selected by the OWSCA algorithm and submitted to the first NMR calculations containing the PC and the Val882 residue.

Subsequently, a second MD simulation (50 ns) was performed between the PC and the water molecules. Similarly, 98 frames were selected by OWSCA to decrease the number of NMR calculations. From the analysis of the Minimum Distance Distribution Function (MDDF) of the water molecules with respect to the PC, it was possible to adopt a radius of 3.2 Å. In this way, all water molecules present within a radius of 3.2 Å were included in the NMR calculation, ensuring that all interactions, specific and non-specific, were considered.

Finally, NMR calculations in two chemical environments (enzymatic and aqueous) were performed at a theory level GIAO–PBEPBE/NMR-DKH, showing chemical shifts of $-1306.58$ ppm and $-2911.25$ ppm, respectively. This change indicates that the $\delta(^{195}\text{Pt})$ is sensitive to the environment change, pointing out that the PC has the potential to be used as a spectroscopic probe.

Overall, this work proposes a strategic path to obtain the chemical shifts in enzymatic and aqueous environments. When starting a study containing a specific FF for the PC, this leads to a more reliable investigation with, therefore, more refined results. In addition, in the selection of the best frames and residues, we achieved a decrease in the quantum calculations and computational cost. Thus, this work inspires new spectroscopic probes to be obtained from specific FFs, allowing deeper and more reliable analyses.

**Author Contributions:** T.M.R.S. and T.C.R. conceived the overarching project; T.M.R.S. performed and analyzed the docking and MD simulations; G.A.A. performed and analyzed NMR calculations; M.A.G. selected the best frames through the OWSCA algorithm; C.A.T. and T.C.R. reviewed the work analyses; C.A.T. and T.C.R. reviewed all manuscript writing. All authors have read and agreed to the published version of the manuscript.

**Funding:** This research was funded by Conselho Nacional de Desenvolvimento Científico e Tecnológico (CNPq), grant number 307837/2014-9, and Fundação de Amparo à Pesquisa do Estado de Minas Gerais (FAPEMIG), grant number PPM-00831-15.

**Institutional Review Board Statement:** Not applicable.

**Informed Consent Statement:** Not applicable.

**Data Availability Statement:** Data associated with this research are available and can be obtained by contacting the corresponding author.

**Acknowledgments:** The authors thank the Brazilian agencies CNPq, FAPEMIG, and CAPES for the financial support.

**Conflicts of Interest:** The authors declare no conflict of interest.

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
