# Peer review of "Improving the Path to Obtain Spectroscopic Parameters for the PI3K—(Platinum Complex) System: Theoretical Evidences for Using 195Pt NMR as a Probe"

_magnetochemistry, doi:10.3390/magnetochemistry9040089_

Round 1

Reviewer 1 Report

The work is fairly executed and written, but I still have some comments.

The paper’s title is too over the top and should reveal work done. Implications of diagnosis and treatment of cancer are pretty speculative. Figure 1 could be improved on the coordination environment as a better representation of the square planar platinum II sphere. Also, “cis” should be in italic on the caption.

As spherical distributions, is the MMDF for Oxygens have two solvation shells? The first is not a genuine shell considering the solute size and the peak’s distance. Is it possible to depict Pt and water interactions, since it is the metal the principal sign they want to improve? The authors could improve this discussion over the structural aspects of the solution. This water arrangement around platin can also help to explain better the difference in the shifts in the enzymatic and water environments.

A typo founded: B3LPY (assuming that is B3LYP instead)

About the references, there is a lot of self-citation, which is a particular concern since authors validate some tools developed independently. Very few references to platinum force-fields previous development also. 

Author Response

Ms. Hana Yang

Assistant Editor

Magnetochemistry

Editor Email: hana.yang@mdpi.com

Dear Ms. Hana Yang,

We very much appreciate the e-mail response containing the reviewers' valuable and helpful feedback on our manuscript. Please find attached the new version including the suggestions made by the Reviewers.

Thus, we performed a complete review of the manuscript with emphasis on scientific content and formatting request. We hope that all necessary improvements from the discussion have been implemented. The list of changes and responses to reviewers' comments are listed below, and modified parts of the article are highlighted in bold.

In addition, we highlight the following three suggestions from the Assistant Editor:

  1. The current words number is 3320 (excluded front matter and
    references in back matter) while the suggested minimum word count is
    4000 words. So please add the maintext words up to 4000.

Author reply: Thank you for your advice, we are sorry for our mistake in the previously version of the manuscript. In this revised version, we believe that our main text has around 4055 words.

  1. Revise the references: There are too many self citations.

Author reply: In this revised version of the manuscript, we added some new and relevant references addressing the reviewers’ suggestions. In addition, we removed 3 (of a previously total of 9) citations from our group works. An explanation for the need of the remain 6 citations was presented in our reply to question 5 of Reviewer #1.

  1. Replace the following emails with institutional emails:

Author reply: Below we provide the institutional e-mails, as requested.

Dr. Taináh Martins Resende Santos  tainah.santos1@estudante.ufla.br
Dr. Gustavo de Almeida Andolpho  gustavo.andolpho4@estudante.ufla.br

Dr. Camila Assis Tavares  camila.tavares1@estudante.ufla.br
Dr. Mateus A Gonçalves  mateusufla@gmail.com (Dr. Gonçalves does not have an institutional e-mail due his post-doc position at UFLA)

Dr. Teodorico de Castro Ramalho teo@ufla.br

Reviewer Comments:

Reviewer 1:

  1. The paper’s title is too over the top and should reveal work done. Implications of diagnosis and treatment of cancer are pretty speculative.

Author reply: Thank you very much for your comments. Because of the pertinent issue raised, we proposed the following title:

“Improving the path to obtain spectroscopic parameters for the PI3K–(platinum complex) system: theoretical evidences for using 195Pt NMR as a probe

  1. Figure 1 could be improved on the coordination environment as a better representation of the square planar platinum II sphere. Also, “cis” should be in italic on the caption.

Author reply: Thank you very much for your comments to improve our work. Based on the suggestions above, we proposed a new Figure 1 that was included in the final version of the paper, as presented below. Finally, we made a correction in the figure caption, considering the spelling of "cis" in italic. Thank you very much.

Figure 1. 3D structure of the cis-dichloro(2-aminomethylpyridine) platinum(II) bonded to 2-(4’-amino2’-hydroxyphenyl)benzothiazole (AHBT).

  1. As spherical distributions, is the MMDF for Oxygens have two solvation shells? The first is not a genuine shell considering the solute size and the peak’s distance. Is it possible to depict Pt and water interactions, since it is the metal the principal sign they want to improve? The authors could improve this discussion over the structural aspects of the solution. This water arrangement around platin can also help to explain better the difference in the shifts in the enzymatic and water environments.

Author reply: We really appreciate your comments. Thank you very much. Indeed, it is possible to observe two solvation shells for the oxygen atoms of the waters, in the MDDF the first peak at ~1.9 Å usually is related to the specific interaction (essentially hydrogen bonds, as discussed in 10.1021/acs.jctc.7b00599 and 10.1016/j.molliq.2021.117945), when a second peak, in our case at ~2.5 Å, is relative to non-specific interactions. It is important to notice that this low peak at ~1.9 Å was expected, once the H-Bond occurs between amide hydrogen or the hydroxyl oxygen of PC and the water molecules. Another important observation to make is that the nitrogen in the aromatic ring must interact mainly with the hydroxyl hydrogen and, to a lesser extent, with water, which can be confirmed by the small contribution of N to MDDF in Figure 7b.

As we already cited in the previously paragraph, we plotted a second graph containing the contributions of all atoms of the complex to the total MDDF (see Figure 7b). As can be seen, the Pt atom has no significant contributions to the total minimum distance distribution function, i.e., the platinum atom does not significantly affect the structure of water molecules, indicating that no significant interactions occur between Pt and water molecules. It is worth noting that other atoms, hydrogen mainly, contribute more significantly to the function of water. It is possible that in smaller platinum compounds, a greater influence of the metal on the MDDF values of the water might be observed.  

The following paragraph and the second graph have been added in the Page 8, as presented below:

Furthermore, from Figure 7b it can be seen that the hydrogen atoms contribute the most to the total function, whereas the Pt atom practically does not contribute to the total minimum distance distribution function (blue curve in Figure 7b).

Figure 7. Minimum Distance Distribution Function of water with respect to PC. (a) Contribution of O (red) and H (green) atoms to the total function (blue). (b) Contribution of the PC atoms to the total function (blue).

It is important to comment that due to suggestions from the reviewers, the order of the figures has changed. Thus, the MDDF analysis that was previously presented in Figure 5 has now become Figure 7.

  1. A typo founded: B3LPY (assuming that is B3LYP instead).

Author reply: We appreciate the attention to detail in our work. In fact, there was a typo in our writing. Our sincere apologies. The correction has been made on line 78 of the manuscript and highlighted in bold. Thank you very much.

“where the B3LYP functional”

  1. About the references, there is a lot of self-citation, which is a particular concern since authors validate some tools developed independently. Very few references to platinum force-fields previous development also.

Author reply: Thank you for your comments. We would like to say that a review of the references used in the manuscript was carried out in order to confirm the importance of each reference to support the information covered. In this sense, we removed three references, in line with the referee´s comment, of the papers from our group cited originally. However, some of the references are essential to the work, namely references [2], [3], [6], [8], [17] and [26].

For instance, reference [2] was important to demonstrate the timeliness of the topic and in the discussion of the NMR results. Reference [3] was one of the inspirations for our paper since this work perform an NMR calculation in a similar platinum complex, but without classical MD simulation.

References [6] and [17] was important to describe the platinum complex force field used and the basis set for NMR calculations, respectively, in our methodology.

Finally, reference [8] the original citation for the OWSCA method and [26] a new and recent version of this theoretical approach. We believe those references are important to discuss the OWSCA results.

We also appreciate the second issue addressed by the reviewer. Very important indeed. So, to address this pertinent observation, we have included in the final version of the manuscript (Page 5) the following discussion, along with the new references cited:

There are reports in the literature about the development of an FF for the cisplatin molecule. These studies began in 1994 through the work of Yao and collaborators 19, where they used statistical analysis to infer on bond stretching and bond angles parameters. This study was further refined later by the work of Scheeff and collaborators 20. The cited works served as a starting point for the formulation of a new methodology for parameterization of coordination compounds with platinum developed from quantum molecular dynamics calculations 21.

In this same perspective, it is possible to find in the literature works that present other parameterization strategies for OPLA-SS2 22 and POSSIM 23 force fields 24. However, in many cases, especially involving metal complexes, the developed parameters are not transferable among several complexes since each compound has its peculiarities. For these reasons, the absence of parameters to describe platinum complexes in MD simulations remains a major challenge. Thus, the development of a specific force field for the PC 6 paved the way for a reliable MD simulation enabling the study of PC with an important biological target, PI3K.

We acknowledge again the reviewers' comments, which have enabled us to significantly improve our paper.

Furthermore, we thank the editorial assistance and hope with the changes and clarifications implemented, the manuscript would be now acceptable for publication in the Magnetochemistry. Finally, we also remain at your disposal for any further inquiries.

With best regards,

Teodorico C. Ramalho

Department of Chemistry,

Federal University of Lavras,

Campus Universitário, C.P. 3037, 37200-000, Lavras,

Brazil.

e-mail:  teo@ufla.br

          http://www.nucleoestudo.ufla.br/gqc/

          https://orcid.org/0000-0002-7324-1353

Reviewer 2 Report

The authors use state of the art computational technics to compute NMR shifts of interest in medicinal chemistry. The article is well written and organized, however there are some methodological gaps that should be filled before publication. Therefore, I suggest the manuscript to be published after the authors address the following issues:

1. Methodology, section 2.1: I think the authors could elaborate more on how they perform the docking simulations. For instance, how do they select the active site of the protein? What is the resolution of the grid? Which residues were considered flexible? Were torsions of the ligand allowed to rotate during docking?

2. The authors say they collected 200 poses from the docking simulation. How did they select the best orientation for MD? The authors should provide a figure or a table with binding energies. Usually dock conformations are clustered according to their binding energies.

3. Figure 2: Did the authors considere the endocyclic N as hydrogen bond acceptor and the amine group coordinated to the Pt metal as hydrogen bond donor?

4. Page 4, lines 150 – 161: The authors show three main hydrogen bonds. Since they use VMD to analyze those interactions, I kindly ask the authors to provide in a supporting information file, or in the main manuscript, the distance vs. time graphs of those hydrogen bonds. For instance, between the heavy atoms involved: NPC23…OVal882, NVal882…OPC48, OPC48…OGlu880. I also suggest the authors to use this notation with subindexes, which is more clear.

5. Figure 6 looks very unprofessional and could be improved. For example, the authors could make both graphs (a and b) of the same size and use the same font size. Figure 6 (a) is bigger than (b), but the former has a smaller font size.

6. I kindly ask the authors to provide a snapshot of the PC:PI3K complex with the amino acids residues that they used for the NMR computations. It is not clear to me if the authors used a reduced model system and if so, how did they perform the cut.

7. Following query 6. How can the authors guarantee that their reduced model system includes all the essential aminoacids that stabilize and destabilize the complex? A missing aminoacids could change the chemical environment and thus give different chemical shifts.

8. Table 1. How did the authors obtain the numbers? Are they average values of the 93 and 98 frames? If that is the case, I think the authors should also provide a measure of deviation.

9. None of the references follow the journal guidelines:

Author 1, A.B.; Author 2, C.D. Title of the article. Abbreviated Journal Name YearVolume, page range. 

Please revise all of them. Also avoid the use of et al. and include all the authors.

Author Response

Ms. Hana Yang

Assistant Editor

Magnetochemistry

Editor Email: hana.yang@mdpi.com

Dear Ms. Hana Yang,

We very much appreciate the e-mail response containing the reviewers' valuable and helpful feedback on our manuscript. Please find attached the new version including the suggestions made by the Reviewers.

Thus, we performed a complete review of the manuscript with emphasis on scientific content and formatting request. We hope that all necessary improvements from the discussion have been implemented. The list of changes and responses to reviewers' comments are listed below, and modified parts of the article are highlighted in bold.

In addition, we highlight the following three suggestions from the Assistant Editor:

  1. The current words number is 3320 (excluded front matter and
    references in back matter) while the suggested minimum word count is
    4000 words. So please add the maintext words up to 4000.

Author reply: Thank you for your advice, we are sorry for our mistake in the previously version of the manuscript. In this revised version, we believe that our main text has around 4055 words.

  1. Revise the references: There are too many self citations.

Author reply: In this revised version of the manuscript, we added some new and relevant references addressing the reviewers’ suggestions. In addition, we removed 3 (of a previously total of 9) citations from our group works. An explanation for the need of the remain 6 citations was presented in our reply to question 5 of Reviewer #1.

  1. Replace the following emails with institutional emails:

Author reply: Below we provide the institutional e-mails, as requested.

Dr. Taináh Martins Resende Santos  tainah.santos1@estudante.ufla.br
Dr. Gustavo de Almeida Andolpho  gustavo.andolpho4@estudante.ufla.br

Dr. Camila Assis Tavares  camila.tavares1@estudante.ufla.br
Dr. Mateus A Gonçalves  mateusufla@gmail.com (Dr. Gonçalves does not have an institutional e-mail due his post-doc position at UFLA)

Dr. Teodorico de Castro Ramalho teo@ufla.br

Reviewer Comments:

Reviewer 2:

  1. Methodology, section 2.1: I think the authors could elaborate more on how they perform the docking simulations. For instance, how do they select the active site of the protein? What is the resolution of the grid? Which residues were considered flexible? Were torsions of the ligand allowed to rotate during docking?

Author reply: Thank you very much for your comments. The crystallographic structure of PI3K and the active ligand (N-{6-[2-(methylsulfanyl)pyrimidin-4-yl]-1,3-benzothiazol-2-yl}acetamide) docked to the active site of the protein is provided from the work of D'Angelo and co-workers (DOI 10.1021/jm1014605). In this way, the position of the active ligand served as a direction to the region of the PI3K active site where we docked the PC.

The resolution of the grid was set from the default value of the Molegro, i.e., 0.3 Å.

As expressed in line 86 of the manuscript, the residues considered flexible were those present within 11 Å of the platinum complex position. By way of clarification, there were a total of 155 flexible residues, namely: LEU661; GLN662; THR746; LYS750; SER753; ASP758; VAL759; SER760; ILE764; TYR787; GLU799; LYS800; CYS801; LYS802; VAL803; MET804; ALA805; SER806; LYS807; LYS808; LYS809; PRO810; LEU811; TRP812; LEU813; GLU814; GLU826; THR827; ILE828; GLY829; ILE830; ILE831; PHE832; LYS833; HIS834; GLY835; ASP836; ASP837; LEU838; ARG839; GLN840; ASP841; MET842; LEU843; ILE844; LEU845; GLN846; ILE847; LEU848; ARG849; MET851; LEU865; PRO866; TYR867; GLY868; CYS869; ILE870; SER871; LYS875; ILE876; GLY877; MET878; ILE879; GLU880; ILE881; VAL882; LYS883; ASP884; ALA885; THR886; THR887; ILE888; ALA889; LYS890; ILE891; GLN892; GLN893; SER894; GLY900; PHE902; TRP910; LYS914; TYR935; CYS936; VAL937; ALA938; THR939; PHE940; VAL941; LEU942; GLY943; ILE944; GLY945; ASP946; ASP947; HIS948; ASN949; ASP950; ASN951; ILE952; MET953; ILE954; THR955; GLU956; THR957; GLY958; ASN959; LEU960; PHE961; HIS962; ILE963; ASP964; PHE965; GLY966; HIS967; GLU981; ARG982; VAL983; PRO984; PHE985; VAL986; LEU987; LEU1030; PHE1031; MET1034; LEU1035; GLY1038; MET1039; PRO1040; SER1044; GLU1046; ASP1047; ILE1048; GLU1094; TYR1050; ILE1051; ARG1052; ASP1053; ALA1054; LEU1055; TYR1067; PHE1068; GLN1071; VAL1074; CYS1075; LYS1078; THR1081; VAL1082; GLN1083; PHE1084; ASN1085; TRP1086; PHE1087; LEU1088; HIS1089.

Finally, we would like to comment that there were no restrictions on the movement of the platinum complex during the docking simulation.

For further clarification and in view of the reviewer's comments, we have reworded the paragraph in lines 81-88, as highlighted in bold below:

The crystallographic structure of PI3K and the active ligand (N-(6-[2-(methylsulfanyl)pyrimidin-4-yl]-1,3-benzothiazol-2-yl) acetamide) docked in the active site of the PI3K, are provided from the work of D'Angelo and collaborators (PDB code: 3QJZ) 9. Thus, the position of the active ligand served as a guide to the PI3K active site region where PC was docked. The simulation was performed considering a constraint of 11 Å and flexible residues at the same distance (totaling 155 flexible residues). In addition, the resolution of the grid used was 0.30 Å. A total of 100 poses were requested in the simulation using the Molegro Virtual Docker software 10.

  1. The authors say they collected 200 poses from the docking simulation. How did they select the best orientation for MD? The authors should provide a figure or a table with binding energies. Usually dock conformations are clustered according to their binding energies.

Author reply: Great question, thank you very much. Indeed, more information about the way we select the best pose is needed. First of all, we would like to point out an error on our part, 100 poses were obtained from the docking simulation and not 200 poses. We apologize. This error has been corrected in the final version of the manuscript.

The main criterion for selecting the best fitting pose was not just energetic but structural. The most important direction was considering the spatial orientation of PC, where the pose chosen presented a conformation closer to the crystallographic structure of the active ligand of PI3K (N-(6-[2-(methylsulfanyl)pyrimidin-4-yl]-1,3-benzothiazol-2-yl) acetamide). Furthermore, the interactions between the PC and PI3K were considered as well, in which the pose selected was able to reproduce the data reported in the literature, H-Bond between PC and Val882 (DOI 10.3390/molecules24213970), with favorable interaction energy, among the 100 poses generated in this investigation.

Finally, to better clarify the criteria for choosing the pose, we have included the following information in bold in the methodology section (lines 89-95), as well as the new Figure 2.

The poses obtained from the docking study were analyzed so that only one could be selected. Among the various possible orientations of the PC in the PI3K active site, we selected the conformation that presented a structure orientation closer to the crystallographic structure of the active ligand of PI3K (N-(6-[2-(methylsulfanyl)pyrimidin-4-yl]-1,3-benzothiazol-2-yl) acetamide) (Figure 2). Furthermore, another relevant criterion was reproducing the interactions already reported in the literature with favorable interaction energy (H-Bonds between PC and residue Val882). In this sense, the most favorable orientation was prepared for the next step, the classical molecular dynamics simulation.”

Figure 2. Overlapping of the selected pose (purple) and the PI3K active ligand (N-(6-[2-(methylsulfanyl)pyrimidin-4-yl]-1,3-benzothiazol-2-yl) acetamide) (yellow).

In addition, we have added the following bold sentence in the results section (lines 161-162):

“After analyzing the 100 poses obtained from the docking calculation, the best orientation was selected (Figure 3). As can be seen, three hydrogen bonds (H-Bonds) were identified involving the platinum complex and the valine (Val882) and glutamic acid (Glu880) amino acids. In addition, the pose interaction energy was – 158.7 kcal/mol, suggesting that PC interacts favorably in the PI3K cavity.

  1. Figure 2: Did the authors considere the endocyclic N as hydrogen bond acceptor and the amine group coordinated to the Pt metal as hydrogen bond donor?

Author reply: Thank you for your comments. We would like to comment that all atoms with possibilities of hydrogen bond formation with the residues of the protein were taken into consideration. However, only N23 and O48 of the platinum complex (Figure 1) showed significant interactions with the protein (Figure 5). In line 224, it is possible to check the percentage frequency of occurrence of hydrogen bonds between these two atoms and the residue Val882.

To improve the discussion raised, we plotted four graphs and present them below. The graphs in Figure 1 refer to the two nitrogen atoms commented on by the reviewer (N30 and N36, Figure 1a e 1b). As can be seen from the first graph, hydrogen bonds begin to appear from 4.0 Å, but still with 5% of occurrence. The same can be analyzed in the second graph, where the H-Bonds begin to form only after 4.5 Å, yet with a very small number of occurrences (0.75%). In this sense, it is possible to infer that the interactions performed between the nitrogen atoms mentioned in the question and the residues of the protein are at long distances, therefore weaker interactions.

(b)

(a)

Figure 1. (a) Hydrogen bonds between N30 and residues of PI3K; and (b) Hydrogen bonds between N36 and residues of PI3K.

On the other hand, as can be observed in Figure 2, where the H-Bonds evaluated refer to the N23 and O48 atoms (Figure 2a e 2b, respectively), it is possible to observe that, in fact, there are pronounced interactions starting at 3 Å. Moreover, with an already significant frequency of occurrence, 62.33% and 15.50%, respectively. Thus, we could conclude that interactions are much more significant than the interactions involving the endocyclic N (N30 and N36).

(b)

(a)

Figure 2. (a) Hydrogen bonds between N23 and residues of PI3K; and (b) Hydrogen bonds between O48 and residues of PI3K.

  1. Page 4, lines 150 – 161: The authors show three main hydrogen bonds. Since they use VMD to analyze those interactions, I kindly ask the authors to provide in a supporting information file, or in the main manuscript, the distance vs. time graphs of those hydrogen bonds. For instance, between the heavy atoms involved: NPC23···OVal882, NVal882···OPC48, OPC48···OGlu880. I also suggest the authors to use this notation with subindexes, which is more clear.

Author reply: Thank you for your comments and suggestions. First, we would like to comment that some figures have changed numbering due to changes made during revision. Thus, we would like to point out that on Page 5 there was the old Figure 2 (now Figure 3) where, indeed, three hydrogen bonds are indicated. However, these H-Bonds refer to the docking study, i.e., there is no associated time scale. On the other hand, on Page 6, Figure 5 is present, where two H-Bonds are indicated. It is worth noting that these two H-Bonds were also indicated in the docking study. Thus, to meet the pertinent suggestion of the reviewer, we plotted the graphs of distances vs. time for these two H-Bonds presented in Figure 5. For this, we included in Page 7 of the final version of the manuscript the following paragraph and, in sequence, the new Figure 6.

In addition, we plotted the graph of the distances versus time of the two H-Bonds pointed out with the highest frequency of occurrence in the considered analysis time (20-50 ns). The first graph (Figure 6a) refers to the PC:H24···O:Val882 interaction, where the average bond length corresponds to 1.99 Å and the standard deviation was only ± 0.28 Å. The second graph (Figure 6b) corresponds to the Val882:H···O48:PC hydrogen bond, with a mean distance of 2.24 Å and a standard deviation of ± 0.22 Å.”

Figure 6. Bond lengths of the two H-bonds with the highest frequency of occurrence in the MD simulation.

  1. Figure 6 looks very unprofessional and could be improved. For example, the authors could make both graphs (a and b) of the same size and use the same font size. Figure 6 (a) is bigger than (b), but the former has a smaller font size.

Author reply: Thank you for your comment. We are very sorry for the poor quality of these figures, and we took the opportunity to improve the quality of this image and make it more professional. It is important to comment that due to suggestions from the reviewers, the order of the figures has changed. Thus, the OWSCA analysis that was previously presented in Figure 6 has now become Figure 8, and is presented in the sequence.

Figure 8. Energies of the systems studied (original and compressed) at each moment. (a) First MD with PC and PI3K; (b) Second MD with PC and water molecules.

  1. I kindly ask the authors to provide a snapshot of the PC:PI3K complex with the amino acids residues that they used for the NMR computations. It is not clear to me if the authors used a reduced model system and if so, how did they perform the cut.

Author reply: We would like to thank the reviewer for their comment. In this line, we provide a figure below (Figure 9) with the PC and the PI3K residue used for the NMR calculations. This reduced model was made by considering the platinum complex and the amino acid residue that perform H-Bond with PC during the entire molecular dynamic simulation.

An explanation of this cut was presented on lines 311-316 of the main text, as well as the Figure 9. A more detailed explanation is presented in the answer to query 7.

Figure 9. Reduced system of PC:PI3K used for NMR calculations, considering the platinum complex and the Val882 residue.

  1. Following query 6. How can the authors guarantee that their reduced model system includes all the essential aminoacids that stabilize and destabilize the complex? A missing aminoacids could change the chemical environment and thus give different chemical shifts.

Author reply: We thank the reviewer for their commentary. As we answer on query 6, due to the high demand for DFT calculations for the complete system, we selected the amino acid residue to perform the chemical shifts calculations. This selection was made to consider residues that interacts via H-Bond with the Platinum Complex after the MD simulation, i.e., Val882.

As your valuable comment highlight, a missing residue in the reduced model can affect the chemical shift result, whit that, we believe that 50 ns of MD simulation can show the H-Bonds that will remain after a long simulation period, as we spoke in lines 236-237. We know that other residues in the PI3K active site can be important to the stabilization of the complex, but, as we highlight in the manuscript, the main result was the gap between the chemical shifts in solvent and enzyme environment, and for this propose, previously studies [2, 3, 27] showed that only the residues close to the PC, i.e., that interacts via H-Bond with the complex, can be considered to this DFT calculation.

This justification was inserted in the main text on lines 311-316 and attached here.

In line with that, we proposed to reduce the PC:PI3K system to be able to perform GIAO-DFT calculations without a very high computational cost. For this reduction, other studies indicate that only the residues of the PI3K active site in the range for perform H-Bonds can be considered for studies of the complex-enzyme relationship 2, 3, 27.  In the PC:PI3K system, the reduced system is composed by the platinum complex and the residue Val882. The representation of the reduced system can be seen in Figure 9.”

  1. Table 1. How did the authors obtain the numbers? Are they average values of the 93 and 98 frames? If that is the case, I think the authors should also provide a measure of deviation.

Author reply: Thank you very much four your comment. Your assumption was correct, the values in Table 1 was the average values of the 93 and 98 frames. In line with that, we rewrote the lines 323-324 of the manuscript to clarify this and added the standard deviation values in the table, as shown in bold below.

Furthermore, we also add a statistical test (Student’s t-test) to show that the average values are statistically different. This new discussion was included in lines 333-341 of the main text. These changes are shown in bold below, as well as in the main text. 

“With this in mind, chemical shifts calculations were performed at theory level PBEPBE/NMR-DKH, using the 93 and 98 frames selected with amino acid residues and water molecules, respectively. The average chemical shifts values for both environments were presented in Table 1. These calculations point out that, as expected, there is a difference when the chemical environment was changed from water to enzymatic of around 1600 ppm.

Table 1. Theoretical 195Pt chemical shifts in different chemical environments performed at theory level PBEPBE/NMR-DKH.

Chemical Environment

δ(195Pt) (ppm)

Enzymatic

- 1306.58 ± 25.33

Aqueous

- 2911.25 ± 9.66

This large difference between the chemical shift in an enzymatic (-1306.58 ppm) and aqueous (-2911.25 ppm) environments is important to understanding the behavior of PC in these two environments.

Since our goal is to verify the difference in the chemical shift value in the different environments, it is important to verify if the mean results of chemical shifts are statistically different from each other. For this we performed a Student's t-test, considering a confidence level of 95%, taking into account that the null hypothesis is that the two means are equal. For our data, the p-value was equal to 0.029, which leads us to rule out the null hypothesis, that is, the mean chemical shift values for the platinum complex in an aqueous and enzymatic environment are statistically different.

The higher chemical shift in the enzyme active site is an expected result, since the active sites have a hydrophobic characteristic, given that H-Bonds are formed between the PC and the PI3K enzyme, highlighting the importance of these interactions for changes in the chemical shift. The same behavior was observed for similar compounds, with Pt and Re, in previously studies 2, 3.”

  1. None of the references follow the journal guidelines: Author 1, A.B.; Author 2, C.D. Title of the article. Abbreviated Journal Name Year, Volume, page range. Please revise all of them. Also avoid the use of et al. and include all the authors.

Author reply: Thank you very much for the alert. Our sincere apologies. We would like to say that the format of the references has been corrected and is in accordance with Magnetochemistry's guidelines.

Author comments: Besides the important changes suggested by Reviewers #1 and #2, we took the opportunity to improve our discussion about the OWSCA selection. Since this is a relative new methodology, we believe that a deeper discussion about it is important and will improve the manuscript with reliable information. This new discussion was included in lines 276-286 of the main text and presented below:

As described in section 2.2.1, we used the Haar wavelet, from the Daubechies family. The selection of the best wavelet is an important step since the wrong choice, i.e., select a wavelet from a family that does not select properly the frames from MD simulation, can impact the statistical data derivate of the system and will affect all the next steps from the work, in our case, the quantum mechanics calculations 26.

It is worth mentioning that this methodology makes the selection of the frames by a compression of the total MD result, using the wavers coming from the system behavior, different from the traditional cluster selection, which groups the frames according to structural similarity. The advantage of this approach comes from the number of statistical values of the system that lead to the selected frames for the following calculations 26.

We acknowledge again the reviewers' comments, which have enabled us to significantly improve our paper.

Furthermore, we thank the editorial assistance and hope with the changes and clarifications implemented, the manuscript would be now acceptable for publication in the Magnetochemistry. Finally, we also remain at your disposal for any further inquiries.

With best regards,

Teodorico C. Ramalho

Department of Chemistry,

Federal University of Lavras,

Campus Universitário, C.P. 3037, 37200-000, Lavras,

Brazil.

e-mail:  teo@ufla.br

          http://www.nucleoestudo.ufla.br/gqc/

          https://orcid.org/0000-0002-7324-1353

Round 2

Reviewer 2 Report

I would like to thank very much the authors for their clear replies.

I think the manuscript is now ready to be accepted as it is.